# Causal Capsules and Tensor Autoencoders

## ABSTRACT

This paper introduces a set of neural network architectures for forward and inverse causal inference that are consistent with capsule theory and implement multilinear (tensor) factor analysis methods. Forward causal inference is addressed with a causal autoencoder-decoder architecture composed of a set of causal capsules that estimate the latent variables representing the constituent factor of data formation, and a tensor-autoencoder that governs the latent variable interaction. A recurrent non-linear causal capsule chain that employs kernel activations computes the optimal linearized subspace for every causal factor, and implements the kernel multilinear principal component analysis or the kernel multilinear independent component analysis. For distributed computation, we break the chain links and each causal representation is computed separately, shuttling causal information between capsules. The causal factor representations may be computed efficiently by restructuring the input into a hierarchy of parts with a set of part-based causal capsules that are "glommed" together to create a part-based hierarchy of causal capsules. Inverse causal inference, the estimation of causes of effects, is addressed with a multilinear projection architecture that inverts the estimated forward causal model and employs a set of observations to constrain the solution set rendering the problem well-posed.

## 1 INTRODUCTION

Neural networks are being employed increasingly in high-stakes application areas, such as face recognition [Taigman et al. (2014); Huang (2012); Sun et al. (2013); Chen et al. (2015); Xiong et al. (2016)], and medical technologies [Kermany et al. (2018); Madani et al. (2018); Topol (2019)]. Developing a set of neural network architectures that are causally explainable is important in developing a trustworthy machine learning, where "A causes B" means "the effect of A is B", a measurable and experimentally repeatable quantity [Holland (1986)].

Forward causal inference models the mechanism of data formation, and estimates the effects of interventions [Pearl (2000); Imbens & Rubin (2015); Spirtes et al. (2000); Vasilescu et al. (2021); Vasilescu & Terzopoulos (2002a; 2005; 2004)]. Unlike, conventional statistics and conventional machine learning that model the observed data distribution, and make predictions about a variable that has been co-observed with another. Inverse causal inference estimates the causes of effects given an estimated forward causal model that is inverted subject to a set of observations that constrain the solution set [Vasilescu (2011); Vasilescu & Terzopoulos (2007)].

There are two conceptual frameworks for causal inference: DAGs or path analysis and potential-outcome. Donald Rubin and his collaborators have advocated the potential outcome approach which framed causal inference as a missing data problem [Imbens (2020)]. Judea Pearl has been advocating *do*-calculus – a directed acyclic graph approach as a mathematical language that he has unified with structural equations and counterfactuals [Bollen & Pearl (2013)]. Judea Pearl's causation ladder [Pearl (2000)] provides a way of thinking about causal discovery, causal reasoning, and decision making. Pearl & Bareinboim (2014); Bareinboim & Pearl (2016) have parameterized the differences between experimental and observational studies based on possible sources of error.

Tensor data analysis is a type of structural equation modeling that has been employed to perform dimensionality reduction, to develop regression models, and to model cause-and-effect, Fig. 1. Tensor factor analysis has been employed in psychometrics [Tucker (1966); Harshman (1970); Carroll & Chang (1970); Bentler & Lee (1979); Kroonenberg & de Leeuw (1980)], econometrics [Kapteyn et al. (1986); Magnus & Neudecker (1988)], chemometrics [Bro (1997); Acar et al. (2014)], signal processing [de Lathauwer (1997; 2008); Cichocki et al. (2009)], computer vision [Vasilescu & Terzopoulos (2002b); Wang & Ahuja (2003)], computer graphics [Vasilescu (2002); Davis & Gao (2003); Vasilescu & Terzopoulos (2004); Vlasic et al. (2005); Hsu et al. (2005)], and machine learning [Vasilescu (2009); Vasilescu & Terzopoulos (2005)]. In machine learning, tensor methods have been effectively employed to reparameterize neural networks. Neural network weights have been organized

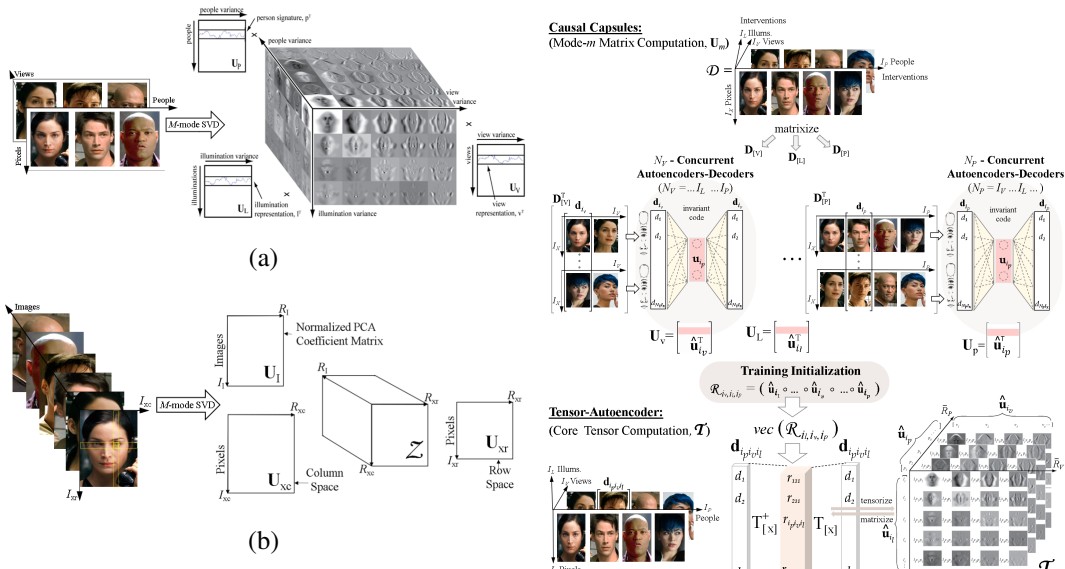

(a)

(b)

Figure 1: (a) $M$-mode SVD estimates the parameters of tensor factor model from a collection of vectorized images that have been acquired combinatorially. (b) $M$-mode SVD computes a regression model and computes the column and row space from a collection of images where each image is a grid of numbers, a "data matrix" or a 2-way array. (All images in this paper have been vectorized, except in this sub-figure.)

Figure 2: Naive neural network implementation of the $M$-mode SVD, alg. 1. Depiction of the TensorFaces model estimation, Fig.1(a). Computing each mode matrix, $\mathbf{U}_m$, naively with a single autoencoder-decoder. The core tensor, $\mathcal{T}$, is computed by an autoencoder that is initialized with the vectorized multilinear (tensor) codes formed from the product of factor representations.

into "data tensors" and dimensionally reduced in order to achieve greater computational efficiency [Lebedev et al. (2014); Novikov et al. (2015); Kim et al. (2015); Khrulkov (2020); Onu et al. (2020); Iwen et al. (2021)]. Tensor methods have also been applied in regression analysis [Kolda et al. (2005); Chu & Ghahramani (2009); Tang et al. (2013); Anandkumar et al. (2014); Kossaifi et al. (2017); Wang et al. (2017); Benesty et al. (2021); Vendrow et al. (2021)].

This paper introduces a set of causal capsule architectures for forward and inverse causal inference that implement tensor factor analysis operations. These architectures are consistent with capsule theory proposed by Geoffrey Hinton and his collaborators [Hinton et al. (2011); Sabour et al. (2017); Hinton (2021)].

Forward causal inference, the estimation of effects of causes, is performed with a causal autoencoder architecture that consists of several causal capsules that compute the causal factor representations, and a tensor-autoencoder that governs the causal factor interaction, Fig 2. A *causal capsule* is formed from a set of constrained "cluster"-based autoencoders[1] that transform the basis vectors spanning the "cluster" subspace, such that a causal factor representation is invariant of the "cluster" membership, *i.e.*, invariant to all the other causal factors [Vasilescu (2009)]. A tensor autoencoder is an autoencoder with a vectorized tensor code formed from the multilinear (tensor) product of factor representations.

A recurrent non-linear causal capsule chain that employs kernel activations computes the optimal linearized subspace for every causal factor, and implements the kernel multilinear principal component analysis or the kernel multilinear independent component analysis, Fig 3. For distributed computation, we break the chain links andeach causal representation is computed separately, shuttling causal information between capsules.

For a scalable architecture, causal representations for an object whole can be computed efficiently by parts, Fig 4. As Hinton (2021) has also indicated, a part-based causal capsule architectures may also be "glommed" together to analyze a hierarchy of data columns [Vasilescu et al. (2021); Vasilescu & Kim (2019); de Lathauwer (2008)]. The hierarchical neural network architecture is a compositional

---

[1]In the context of multifactor data analysis, a cluster is a set of observations for which all factors are fixed except one. Data belonging to the same cluster may not form a cluster in Euclidean space and not easily identifiable by an EM algorithm [ Dempster et al. (1977)].

hierarchical computation of causal factor representation and implements the Incremental $M$-mode Block SVD algorithm. [2]

Inverse causal inference is performed with a multilinear projection architecture [Vasilescu & Terzopoulos (2007); Vasilescu (2009)] that is performed by inverting an estimated forward model subject to data constraints. Fig. 5.

The architectures are derived based on two mathematical principles: (i)linear autoencoders-decoders weights are the principal component analysis basis vectors, sec. 2, (ii) the object-whole representation can be derived bottom-up in closed form from a part-based hierarchical causal factor representation, sec 3.2.

After reviewing the mathematical foundations of our work in the next section, we discuss forward causal models and depict their neural network architectures in Section 3 and discuss inverse causal inference and depict the multilinear projection neural network architectures in Section 4. Section 5 concludes the paper.

---

**Algorithm 1** $M$-mode SVD algorithm.

---

**Input** the data tensor $\mathcal{D} \in \mathbb{C}^{I_0 \times \cdots \times I_M}$.

1. For $m := 0, \ldots, M$,
   Let $\mathbf{U}_m$ be the left orthonormal matrix of $[\mathbf{U}_{\mathrm{m}}\mathbf{S}_{\mathrm{m}}\mathbf{V}_{\mathrm{m}}^{\mathrm{T}}] := \mathrm{svd}(\mathbf{D}_{[m]})$[a]
2. Set $\mathcal{Z} := \mathcal{D} \times_0 \mathbf{U}_0^{\mathrm{T}} \times_1 \mathbf{U}_1^{\mathrm{T}} \cdots \times_m \mathbf{U}_m^{\mathrm{T}} \cdots \times_M \mathbf{U}_M^{\mathrm{T}}$.

**Output** mode matrices $\mathbf{U}_0, \mathbf{U}_1 \ldots, \mathbf{U}_M$ and the core tensor $\mathcal{Z}$.

---

[a]The computation of $\mathbf{U}_m$ in the SVD $\mathbf{D}_{[m]} = \mathbf{U}_m \mathbf{\Sigma} \mathbf{V}_m^{\mathrm{T}}$ can be performed efficiently, depending on which dimension of $\mathbf{D}_{[m]}$ is smaller, by decomposing either $\mathbf{D}_{[m]}\mathbf{D}_{[m]}^{\mathrm{T}} = \mathbf{U}_m \mathbf{\Sigma}^2 \mathbf{U}_m^{\mathrm{T}}$ (note that $\mathbf{V}_m^{\mathrm{T}} = \mathbf{\Sigma}^+ \mathbf{U}_m^{\mathrm{T}} \mathbf{D}_{[m]}$) or by decomposing $\mathbf{D}_{[m]}^{\mathrm{T}}\mathbf{D}_{[m]} = \mathbf{V}_m \mathbf{\Sigma}^2 \mathbf{V}_m^{\mathrm{T}}$ and then computing $\mathbf{U}_m = \mathbf{D}_{[m]}\mathbf{V}_m\mathbf{\Sigma}^+$.

---

## 2 LINEAR AUTOENCODER AND LINEAR PCA

An autoencoder-decoder that minimizes the reconstruction loss function,

$$l = \sum_{i=1}^{I} \|\mathbf{d}_{\mathrm{i}} - \mathbf{B}\mathbf{c}_{\mathrm{i}}\| + \lambda\|\mathbf{B}^{\mathrm{T}}\mathbf{B} - \mathbf{I}\| \tag{1}$$

and has a linear decoder learns a set of weights, $\mathbf{b}_{\mathrm{r}}$ that are identical to the PCA basis vectors when the weights of each neuron, $c_r$ are computed sequentially. An autoencoder is implemented with a cascade of Hebb neurons [Hebb (1949)]. The contribution of each neuron, $c_1, \ldots, c_{\mathrm{r}}$, are the PCA sequentially computed and subtracted from a centered training data set, and the difference is driven through the next Hebb neuron, $c_{\mathrm{r+1}}$ [Sejnowski et al. (1989); Sanger (1989); Rumelhart et al. (1986); Ackley et al. (1985); Oja (1982)]. The weights of a Hebb neuron, $c_{\mathrm{r}}$, are updated by

$$\Delta\mathbf{b}_{\mathrm{r}}(t+1) = \eta\left(\mathbf{d} - \sum_{i_{\mathrm{r}}=1}^{r}\mathbf{b}_{i_{\mathrm{r}}}(t)c_{i_{\mathrm{r}}}(t)\right)c_{\mathrm{r}}(t) = \eta\left(\mathbf{d} - \sum_{i_{\mathrm{r}}=1}^{r}\mathbf{b}_{i_{\mathrm{r}}}(t)\mathbf{b}_{i_{\mathrm{r}}}^{\mathrm{T}}(t)\mathbf{d}\right)\mathbf{d}^{\mathrm{T}}\mathbf{b}_r(t),$$

$$\mathbf{b}_{\mathrm{r}}(t+1) = \frac{(\mathbf{b}_{\mathrm{r}}(t) + \Delta\mathbf{b}_{\mathrm{r}}(t+1))}{\|\mathbf{b}_{\mathrm{r}}(t) + \Delta\mathbf{b}_{\mathrm{r}}(t+1)\|}$$

where $\mathbf{d} \in \mathbb{C}^{I_0}$ is a vectorized centered observation with $I_0$ measurements, $\eta$ is the learning rate, $\mathbf{b}_{\mathrm{r}}$ are the autoencoder weights of the $r$ neuron, $c_{\mathrm{r}}$ is the activation, and $t$ is the time iteration. Backpropagation [LeCun et al. (1988; 2012)] is equivalent to performing PCA gradient descent [Jolliffe (1986)].

## 3 CAUSAL INFERENCE

Throughout this article, we will denote scalars by lower case italic letters $(a, b, ...)$, vectors by bold lower case letters $(\mathbf{a}, \mathbf{b}, ...)$, matrices by bold uppercase letters $(\mathbf{A}, \mathbf{B}, ...)$, and higher-order tensors by bold uppercase calligraphic letters $(\mathcal{A}, \mathcal{B}, ...)$. Index upper bounds are denoted by italic uppercase letters (*i.e.*, $1 \leq a \leq A$ or $1 \leq i \leq I$). The zero matrix is denoted by $\mathbf{0}$, and the identity matrix is denoted by $\mathbf{I}$. The TensorFaces paper [Vasilescu & Terzopoulos (2002a) is a gentle introduction to tensor factor analysis, Kolda and Bader [Kolda & Bader (2009) is a nice survey of tensor methods and references [Vasilescu (2009); de Lathauwer (1997); Bro (1997) provide an in depth treatment of tensor factor analysis.

---

[2]By comparison, a hierarchical Tucker is a resource efficient hierarchical computational scheme that employs a hierarchical re-balancing of the modes trick in which one flattens a data tensor in multiple modes at the same time to avoid computing SVDs of skinny matrices [Hackbusch & Kühn (2009); Grasedyck (2010); Perros et al. (2015)].

---

**Algorithm 2** Kernel Multilinear PCA/ICA (K-MPCA/MICA) algorithm.

---

**Input** the data tensor $\mathcal{D} \in \mathbb{C}^{I_0 \times \cdots \times I_M}$, where mode $m = 0$ is the measurement mode, and the desired ranks $\tilde{R}_1, \ldots, \tilde{R}_M$.

    1. For $m := 1, \ldots, M$,

        Compute the elements of the mode-$m$ covariance matrix, for $j, k := 1, \ldots, I_m$, as follows:

$$[\mathbf{D}_{[m]}\mathbf{D}_{[m]}{}^{\mathrm{T}}]_{jk} := \sum_{i_1=1}^{I_1} \ldots \sum_{i_{m-1}=1}^{I_{m-1}} \sum_{i_{m+1}=1}^{I_{m+1}} \ldots \sum_{i_M=1}^{I_M} K(\mathbf{d}_{i_1 \ldots i_{m-1}\, j\, i_{m+1} \ldots i_M}, \mathbf{d}_{i_1 \ldots i_{m-1}\, k\, i_{m+1} \ldots i_M}).$$

$\Bigg\{$

    *For K-MPCA:*   Set $\mathbf{U}_m$ to the left matrix of the SVD of $\mathbf{D}_{[m]}\mathbf{D}_{[m]}{}^{\mathrm{T}} = \mathbf{U}_m \mathbf{\Sigma}^2 \mathbf{U}_m{}^{\mathrm{T}}$
                    Truncate to $\tilde{R}_m$ columns $\mathbf{U}_m \in \mathbb{C}^{I_m \times \tilde{R}_m}$.

    *For K-MICA:*   Compute $\mathbf{U}_m := \mathbf{C}_m \in \mathbb{C}^{I_m \times \tilde{R}_m}$ based on
                    [Vasilescu & Terzopoulos (2005)]. The initial SVD truncates to $\tilde{R}_m$.

    2. Set $\mathcal{T} := \mathcal{D} \times_1 \mathbf{U}_1^+ \cdots \times_m \mathbf{U}_m^+ \cdots \times_M \mathbf{U}_M^+$.

    3. *Local optimization via alternating least squares:*

        Iterate for $n := 1, \ldots, N$

            For $m := 1, \ldots, M$,

                Set $\mathcal{X}_m := \mathcal{D} \times_1 \mathbf{U}_1^+ \cdots \times_{m-1} \mathbf{U}_{m-1}^+ \times_{m+1} \mathbf{U}_{m+1}^+ \cdots \times_M \mathbf{U}_M^+$.

                Set $\mathbf{U}_m$ to the $\tilde{R}_m$ leading left-singular vectors of the SVD of $\mathbf{X}_{m,[m]}{}^{a}$.

            Set $\mathcal{T} := \mathcal{X}_M \times_M \mathbf{U}_M^+$.

        until convergence.

**Output** the converged extended core tensor $\mathcal{T} \in \mathbb{C}^{I_0 \times \tilde{R}_1 \times \cdots \times \tilde{R}_M}$ and causal factor mode matrices $\mathbf{U}_1, \ldots, \mathbf{U}_M$.

---

$^a$See Alg. 1, footnote $a$

| | |
|---|---|
| Linear kernel: | $K(\mathbf{u}, \mathbf{v}) = \mathbf{u}^{\mathrm{T}}\mathbf{v} = \mathbf{u} \cdot \mathbf{v}$ |
| Polynomial kernel of degree $d$: | $K(\mathbf{u}, \mathbf{v}) = (\mathbf{u}^{\mathrm{T}}\mathbf{v})^d$ |
| Polynomial kernel up to degree $d$: | $K(\mathbf{u}, \mathbf{v}) = (\mathbf{u}^{\mathrm{T}}\mathbf{v} + 1)^d$ |
| Sigmoidal kernel: | $K(\mathbf{u}, \mathbf{v}) = \tanh(\alpha \mathbf{u}^{\mathrm{T}}\mathbf{v} + \beta)$ |
| Gaussian (radial basis function (RBF)) kernel: | $K(\mathbf{u}, \mathbf{v}) = \exp\left(-\frac{\|\mathbf{u}-\mathbf{v}\|^2}{2\sigma^2}\right)$ |

Table 1: Common kernel functions. Kernel functions are symmetric, positive semi-definite functions(corresponding to symmetric, positive semi-definite Gram matrices). The linear kernel does not modify or warp the feature space.

### 3.1 FORWARD CAUSAL INFERENCE

Forward causal inference frames questions in terms of interventions. What if the causal factor $c$ were changed by one unit, how much would the observed measurements, $\mathbf{d}$, be expected to change?

For modeling individual level-effects rather than the average effects of causes, observations are acquired by systematically varying each causal factor while holding the rest of the causal factors fixed.

Within the tensor mathematical framework, a $M$-way array or "data-tensor", $\mathcal{D} \in \mathbb{C}^{I_0 \times I_1 \cdots \times I_m \cdots \times I_M}$ contains a collection of vectorized and centered observations,[3] $\mathbf{d}_{i_1 \ldots i_m \ldots i_M} \in \mathbb{R}^{I_0}$ that are the result of $M$ causal factors. The $m$ causal factor ($1 \leq m \leq M$) takes one of $I_m$ values that are indexed by $i_m, 1 \leq i_m \leq I_m$. An observation that is result of the confluence $M$ causal factors is modeled by a multilinear tensor equation with multimode latent variables,

$$\mathbf{d}_{i_1, \ldots, i_M} = \mathcal{T} \times_1 (\mathbf{r}_{i_1}^{\mathrm{T}} + \epsilon_{i_1}^{\mathrm{T}}) \cdots \times_M (\mathbf{r}_{i_M}^{\mathrm{T}} + \epsilon_{i_M}^{\mathrm{T}}) + \epsilon_{i_1, \ldots, i_M}$$

where $\mathcal{T} = \mathcal{Z} \times_0 \mathbf{U}_0$ is the extended core which modulates the interaction between the latent variables, $\mathbf{r}_{i_1} \ldots \mathbf{r}_{i_m} \ldots \mathbf{r}_{i_M}$, that represent the causal factors, and $\boldsymbol{\epsilon}_{i_m} \in \mathcal{N}(\mathbf{0}, \mathbf{\Sigma}_m)$ is an IID Gaussian noise.

---

[3] Reference [(Vasilescu, 2009, Appendix A)] evaluates some of the arguments found in highly cited publications in favor of treating an image as a matrix (tensor) rather than a vector. While technically speaking, it is not incorrect to treat an image as a matrix in linear/tensor algebra, most arguments do not stand up to analytical scrutiny, and it is preferable to vectorize an image and treat it as a single observation rather than a collection of independent column/row observations.

The $M$-mode SVD, Alg. 1 and Alg. 2, and their neural network architecture counterparts, Fig. 2, Fig. 3 may be employed to represent data in terms of their causal factors by minimizing the reconstruction loss function,

$$l = \|\mathcal{D} - \mathcal{T} \times_1 \mathbf{U}_1 \cdots \times_m \mathbf{U}_m \cdots \times_M \mathbf{U}_M\| + \sum_{m=1}^{M} \lambda_m \|\mathbf{U}_m \mathbf{U}_m^T - \mathbf{I}\|, \tag{2}$$

where $\mathcal{T}$ is the extended core and the mode matrices, $\mathbf{U}_m$, spans the $m$ causal factor representation. Each mode matrix, $\mathbf{U}_m$ is computed using alternating least squares where a set of $M$ least squares are computed by moving the mode matrices $\mathbf{U}_1, \ldots, \mathbf{U}_{m-1}, \mathbf{U}_{m+1}, \ldots, \mathbf{U}_M$ to the knowns side of the equation, setting $\mathcal{X}_m := \mathcal{D} \times_1 \cdots \times_{m-1} \mathbf{U}_{m-1}^T \times_{m+1} \mathbf{U}_{m+1}^T \cdots \times_M \mathbf{U}_M^T$ for distributed computation or setting $\mathcal{X}_m := (\mathcal{X}_{m-1} \times_{m-1} \mathbf{U}_{m-1}^T) \times_m \mathbf{U}_m)$ for sequential computation, and optimizing the loss function

$$l = \|\mathcal{X}_m - \mathcal{T} \times_m \mathbf{U}_m\| + \lambda \|\mathbf{U}_m \mathbf{U}_m^T - \mathbf{I}\| = \|\mathbf{X}_{m[m]} - \mathbf{U}_m \mathbf{T}_{[m]}\| + \lambda \|\mathbf{U}_m \mathbf{U}_m^T - \mathbf{I}\|, \tag{3}$$

$$\text{where} \quad \mathcal{X}_m := \mathcal{D} \times_1 \cdots \times_{m-1} \mathbf{U}_{m-1}^T \times_{m+1} \mathbf{U}_{m+1}^T \cdots \times_M \mathbf{U}_M^T \quad \text{- distributed computation} \tag{4}$$

$$= (\mathcal{X}_{m-1} \times_{m-1} \mathbf{U}_{m-1}^T) \times_m \mathbf{U}_m = \mathcal{T} \times_m \mathbf{U}_m \quad \text{- sequential computation} \tag{5}$$

The mode matrix $\mathbf{U}_m$ is set to the subspace of the matrixized $\mathcal{X}_m$, $\mathbf{U}_m \mathbf{S}_m \mathbf{V}_m^T := \text{svd}(\mathbf{X}_{m[m]})$[4]. Figure 3(e) displays a recurrent causal chain that unrolls the for-loop from step 3, Alg 2 and sequentially computes the mode matrices by performing an SVD on the a matrixized $\mathcal{X}_m$ computed from eq.( 5). For a distributed computation, $M$ different $\mathcal{X}_m$ are computed according to eq.( 4), where mode matrices are shuttled between the different threads.

## 3.2 Derivation: Invariance and Hierarchy of Causal Capsules

In this section, takes advantages of the principle that an SVD can be computed from its parts. On that basis, we derive a causal factor representation that is statistical invariant to "cluster" membership (*i.e.*, all other causal factors). On that basis, we provide a scalable architecture by deriving a compositional bottom-up computation of an object whole representation. Thus, the naive and direct implementation of the M-mode SVD is replaced with a compositional hierachical part-based distributed architecture.

Computing the mode matrices, $\mathbf{U}_m$, may be viewed as equivalent to computing a set of mutually constrained, cluster-based PCAs. When dealing with data that can be separated into clusters, the standard machine learning approach is to compute a separate PCA. When data from different clusters are generated by the same underlying process (e.g., facial images of the same people under different viewing conditions), the underlying data can be concatenated in the measurement mode and the common causal factor can be modeled by one PCA.[5]

Thus, we define a *constrained, cluster-based PCA* as the computation of a set of PCA basis vectors that are rotated such that the latent representation is constrained to be the invariant of the cluster.

MPCA performs $M$ constrained, cluster-based PCAs, since the computation of the mode-$m$ matrix $\mathbf{U}_m$, which involves a mode-$m$ data tensor flattening and subsequent SVD, is equivalent to performing a constrained, cluster-based PCA; i.e., data cluster concatenation followed by an SVD. This is self evident when employing our modified datum-centric flattening operator, Fig. 7.

In the context of our multifactor data analysis, we define a cluster as a set of observations for which all factors are fixed except one, the $m$ factor. Note that there are $N_m = I_1 I_2 \ldots I_{m-1} I_{m+1} \ldots I_M$ possible clusters and the data in each cluster varies with the same causal mode.[6] Thus, the data across different clusters share one of the underlying causal factors. The constrained, cluster-based PCA concatenates the clusters in the measurement mode and analyzes the data with a linear model, such as PCA or ICA [Bartlett et al. (2002); Common (1994); Lathauwer et al. (1995); De Lathauwer et al. (1996); Anandkumar et al. (2014).

To see this, let $\mathcal{D}_{i_1 \ldots i_{m-1} i_{m+1} \ldots i_M} \in \mathbb{C}^{I_0 \times 1 \times 1 \cdots \times 1 \times I_m \times 1 \cdots \times 1}$ denote a subtensor of $\mathcal{D}$ that is obtained by fixing all modes except causal factor mode $m$ and mode (the measurement mode). Matrixizing this subtensor in the measurement mode 0, we obtain $\mathbf{D}_{i_1 \ldots i_{m-1} i_{m+1} \ldots i_M [0]} \in \mathbb{C}^{I_0 \times I_m}$. This data matrix comprises a cluster of data obtained by varying the $m$ causal factor, to which one can traditionally

---

[4]See Alg. 1, footnote *a*

[5]The active appearance model concatenated two different measurements, facial feature locations and texture, to compute a person representation for a particular viewpoint invariant of measurement. More generally, one can concatenate the measurements of a person from different viewpoint clusters to compute a person representation that is invariant of the measurement and the viewpoint [Cootes et al. (2001); Si et al. (2013)].

[6]Observations in the cluster may not be in Euclidean proximity in the measurement space. Consequently, a cluster may not be easily identified through a standard EM algorithm.

apply PCA. Since there are $N_m = I_1 I_2 \ldots I_{m-1} I_{m+1} \ldots I_M$ possible clusters that share the same underlying space associated with the $cmth$ factor, the data can be concatenated and PCA performed in order to extract the same representation for the $m$ factor regardless of the cluster. Now, consider the MPCA computation of mode matrix $\mathbf{U}_m$, Fig. 3(a), which can be written in terms of matrixized subtensors as

$$
\mathbf{D_m}^{\mathrm{T}} = \begin{bmatrix} \mathbf{D}_{1\ldots11\ldots1\,[m]}{}^{\mathrm{T}} \\ \vdots \\ \mathbf{D}_{I_1\ldots11\ldots1\,[m]}{}^{\mathrm{T}} \\ \vdots \\ \mathbf{D}_{I_1\ldots I_{m-1}I_{m+1}\ldots I_M\,[m]}{}^{\mathrm{T}} \end{bmatrix} = \mathbf{V}_m \boldsymbol{\Sigma}_m \mathbf{U}_m{}^{\mathrm{T}}. \tag{6}
$$

Clearly, this is equivalent to computing a set of $N_m = I_1 I_2 \ldots I_{m-1} I_{m+1} \ldots I_M$ cluster-based PCAs concurrently by combining them into a single statistical model and representing the underlying causal factor $m$ common to the clusters. Thus, rather than computing a separate linear PCA model for each cluster, MPCA concatenates the clusters into a single statistical model and computes a representation (coefficient vector) for mode $m$ that is invariant relative to the other causal factor modes $1, ..., (m-1), (m+1), ..., M$. Thus, MPCA is a multilinear, constrained, cluster-based PCA.

To clarify the relationship, let us number each of the matrices $\mathbf{D}_{i_1 \ldots i_{m-1} i_{m+1} \ldots i_M\,[m]} = \mathbf{D}_m^{(n)}$ with a parenthetical superscript $1 \leq n = 1 + \sum_{k=1, k \neq m}^{M} (i_n - 1) \prod_{l=1, l \neq m}^{k-1} I_l \leq N_m$.

Let each of the linear SVDs be

$$
\mathbf{D}_m^{(n)} = \mathbf{U}_m^{(n)} \boldsymbol{\Sigma}_m^{(n)} \mathbf{U}_0^{(n)\mathrm{T}} \tag{7}
$$

$$
\tag{8}
$$

$$
\mathbf{D}_{[m]} = \underbrace{\begin{bmatrix} \mathbf{U}_m^{(1)} \boldsymbol{\Sigma}_m^{(1)} & \ldots & \mathbf{U}_m^{(N_M)} \boldsymbol{\Sigma}_m^{(N_m)} \end{bmatrix}}_{\mathrm{SVD}} \mathrm{diag}([\begin{array}{ccc} \mathbf{U}_0^{(1)} & \ldots & \mathbf{U}_0^{(N_m)} \end{array}])^{\mathrm{T}}, \tag{9}
$$

$$
= \mathbf{U}_m \boldsymbol{\Sigma}_m \mathbf{W}_\mathrm{m}^{\mathrm{T}} \mathrm{diag}([\begin{array}{ccc} \mathbf{U}_0^{(1)} & \ldots & \mathbf{U}_0^{(N_m)} \end{array}])^{\mathrm{T}}, \tag{10}
$$

$$
= \mathbf{U}_m \boldsymbol{\Sigma}_m [\begin{array}{ccc} \mathbf{U}_0^{(1)} \mathbf{W}_\mathrm{m}^{(1)} & \ldots & \mathbf{U}_\mathrm{m}^{(N_m)} \mathbf{W}_\mathrm{m}^{(N_m)} \end{array}]^{\mathrm{T}} \tag{11}
$$

where $\mathrm{diag}(\cdot)$ denotes a diagonal matrix whose elements are each of the elements of its vector argument. The mode matrix $\mathbf{U}_0^{(n_\mathrm{m})}$ is the measurement matrix that contains the eigenvectors that span the observed data in cluster $n_\mathrm{m}$, $1 \leq n_\mathrm{m} \leq N_\mathrm{m}$. MPCA can be thought as computing a rotation matrix, $\mathbf{W}_\mathrm{m}$, that contains a set of blocks $\mathbf{W}_\mathrm{m}^{(n)}$ along the diagonal that transform the PCA cluster eigenvectors, $\mathbf{U}_0^{(n_\mathrm{m})}$, such that the mode matrix $\mathbf{U}_m$ is the same regardless of cluster membership, eqs.(9-11). The constrained "cluster"-based PCAs may also be implemented with a set of concurrent "cluster"-based PCAs.

Object wholes appear to have have a hierarchy of perceptual parts. Part-based causal capsule architectures may be "glommed" together to create a hierarchy of part-based causal capsules that analyze a hierarchy of data columns [Vasilescu et al. (2021); Vasilescu & Kim (2019); de Lathauwer (2008)]. This hierarchical architecture implements the incremental hierarchical multilinear (tensor) block decomposition algorithm.

Causal factors of object wholes may be computed efficiently from their parts, Fig 4. The matrixized data tensor may be organized into part "clusters" by applying a permutation

$$
\mathcal{D}^{\mathrm{T}} \times_\mathrm{m} \mathbf{P} \Leftrightarrow \mathbf{P} \mathbf{D}_{[m]}{}^{\mathrm{T}} \tag{12}
$$

where $\mathbf{P}$ is a permutation matrix. The resulting hierarchical architecture implements the Incremental M-mode Block SVD algorithm. The Incremental M-mode Block SVD is a generalized hierarchical part-based decomposition that computes an exact global decomposition and is suitable for streaming data [Vasilescu et al. (2021).

## 3.3 NONLINEAR CAUSAL CAPSULES AND KERNEL MPCA/KERNEL MICA

An autoencoder with a non-linear activation function represents an observation with

$$
\mathbf{d}_\mathrm{i} = f_\mathrm{d}(\mathbf{B}_\mathrm{d} \underbrace{f_\mathrm{e}(\mathbf{B}_\mathrm{e} d_\mathrm{i} + \mathbf{a}_\mathrm{e})}_{\mathbf{c}_\mathrm{i}} + \mathbf{a}_\mathrm{d}), \tag{13}
$$

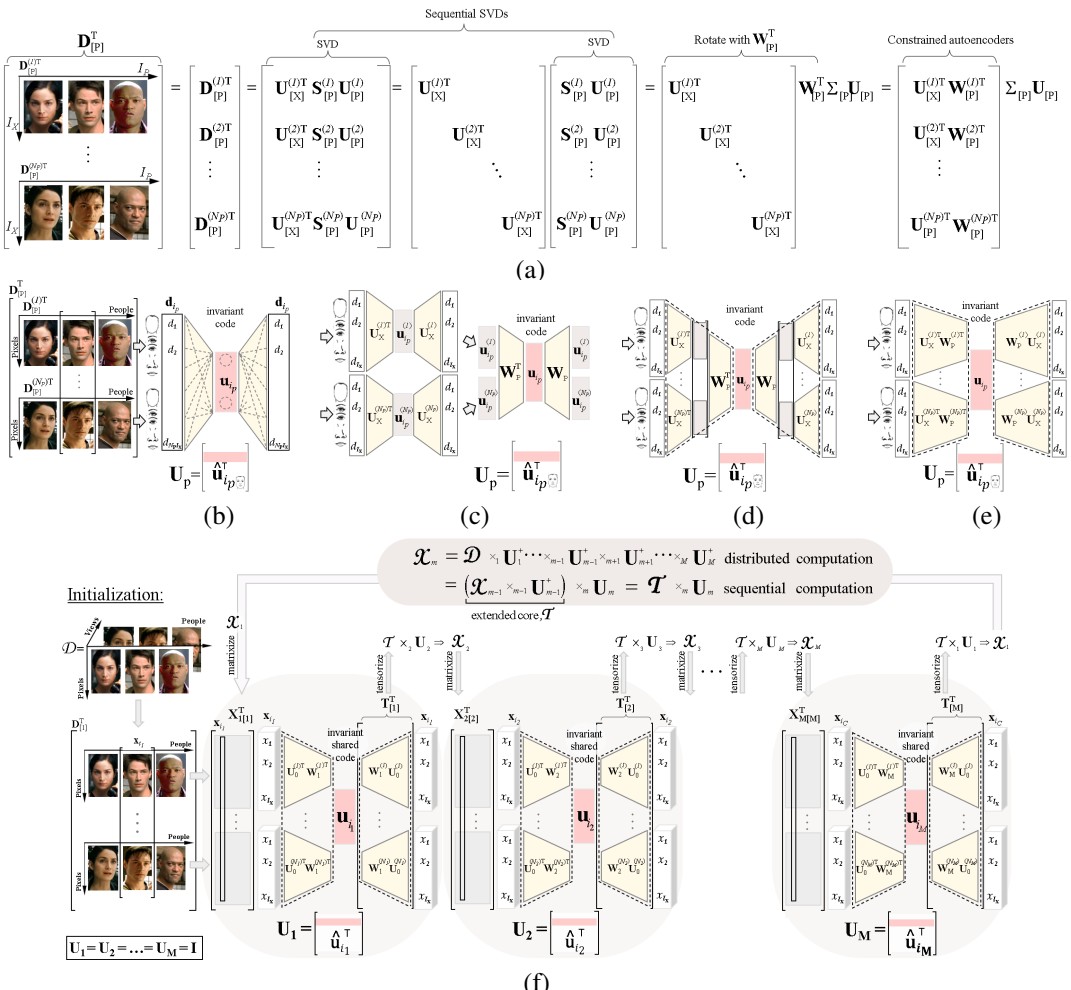

Figure 3: Face recognition example. (a) An ensemble of vectorized images is organized into $\mathcal{D} \in \mathbb{C}^{I_X \times I_P \times I_V \times I_L \times I_E}$ is matrixized into a data matrix, $\mathbf{D}_P$ from which one can compute the mode matrix, $\mathbf{U}_P$, that spans the person representation. This depicts how a single SVD($\mathbf{D}_{[P]}$ can be written in terms of (i) constrained cluster-based autoencoder (PCA) and (ii) concurrent autoencoder. This is depicted as a neural network architecture in (b), (c) and (d), respectively. (b) Mode matrix computation using a single autoencoder-decoder. (c) Mode matrix computation using a constrained cluster-based autoencoder-decoder based on the derivation in part (a). (d) Concurrent-autoencoder. (e) The neural network architecture consists of a chain of constrained autoencoders-decoders where the weights of one constrained autoencoder-decoder are the inputs of the next one. This constrained recurrent causal chain is the unrolled for-loop that computes the mode matrices by employing alternating least squares. When the autoencoders employ kernels then the architecture implements K-MPCA/ K-MICA, Alg. 2.

where $f_e$, $f_d$ are the encoder, decoder activation functions, and $\mathbf{B}_e, \mathbf{B}_d$ are the encoder, decoder weights respectively. Kernel PCA (KPCA) is often given as an example of a "true" nonlinear model. KPCA first applies a nonlinear transformation to the data and then it performs a linear decomposition. Thus, KPCA derives its nonlinearity from its preprocessing step. Other nonlinear methods include nonlinear PCA (NLPCA) [Kramer (1991)], as well as kernel PCA (KPCA) [Schölkoph et al. (1998)] and kernel LDA (KLDA) [Yang (2002)] methods in which kernel functions that satisfy Mercer's theorem correspond to inner products in infinite-dimensional space. An alternative approach is to apply linear models to nonlinear problems through the "kernel trick", specifically the kernel PCA [Schölkoph et al. (1998)] and kernel ICA [Yang et al. (2005)] techniques.[7] Kernel PCA/ICA are

---

[7]The so-called "kernel trick" maps the original non-linear measurements into a higher-dimensional space, where a linear classifier is subsequently used; this makes a linear classification in the new space equivalent to non-linear classification in the original space. This is done using Mercer's theorem, which states that any continuous, symmetric, positive semi-definite kernel $K(\mathbf{u}, \mathbf{v})$ can be expressed as an inner product in a high-dimensional space. Wherever an inner product between two vectors is used, it is replaced with a kernel of the

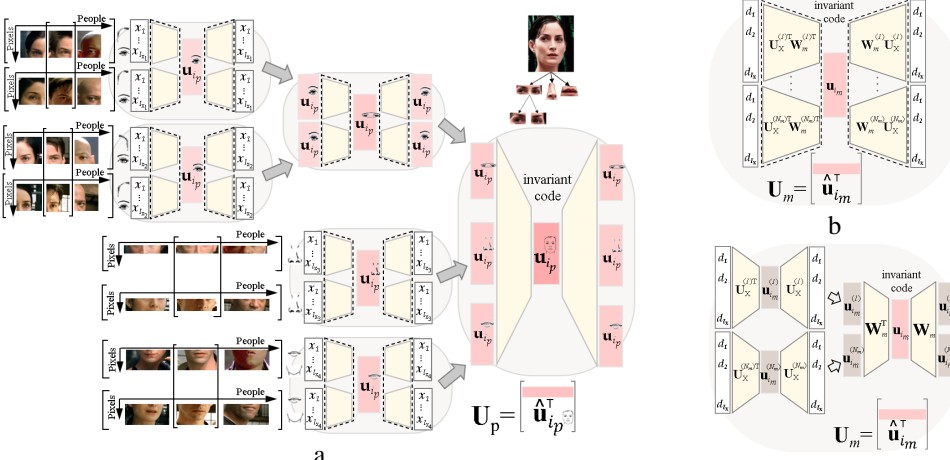

Figure 4: (a) Learning levels of abstractions bottom-up with a hierarchy causal part-based capsules. Each causal capsule in Fig 3 (f) can be replaced with a hierarchy of parts and wholes. The constrained cluster-based PCA (b), may be computed with a hierarchy of autoencoders, as derived in eq. (6)-(10), and depicted in Fig 3.

nonlinear versions of their conventional linear counterparts, but of course they are not multimodal factor models.

The kernel trick can also be applied to our multilinear, multifactor PCA/ICA models to further nonlinearize them, thus enabling them to deal with arbitrarily nonlinear data, Alg. 2.

To accomplish this, recall that the computation of the mode-$m$ covariance matrix $\mathbf{D}_{[m]}\mathbf{D}_{[m]}{}^{\mathsf{T}}$ involves inner products $\mathbf{d}_{i_1\ldots i_{m\text{-}1}\,j\,i_{m+1}\ldots i_{\mathsf{M}}}^{\mathsf{T}}\mathbf{d}_{i_2\ldots i_{m\text{-}1}\,k\,i_{m+1}\ldots i_{\mathsf{M}}}$ between pairs of images in the image data tensor $\mathcal{D}$ associated with causal factor mode $m$, for $m = 1,\ldots,M$. We replace the inner products with a generalized distance measure between images, $K(\mathbf{d}_{i_1\ldots i_{m-1}\,j\,i_{m+1}\ldots i_M}, \mathbf{d}_{i_2\ldots i_{m-1}\,k\,i_{m+1}\ldots i_M})$, where $K(\cdot,\cdot)$ is a suitable kernel function (Table 1), which corresponds to an inner product in some expanded feature space. This generalization naturally leads us to a *Kernel Multilinear PCA (K-MPCA) Algorithm*, where the covariance step computation in in Algorithm 1 is replaced by

$$[\mathbf{D}_{[m]}\mathbf{D}_{[m]}{}^{\mathsf{T}}]_{jk} := \sum_{i_1=1}^{I_1}\cdots\sum_{i_{m-1}=1}^{I_{m-1}}\sum_{i_{m+1}=1}^{I_{m+1}}\cdots\sum_{i_M=1}^{I_M} K(\mathbf{d}_{i_1\ldots i_{m-1}\,j\,i_{m+1}\ldots i_M}, \mathbf{d}_{i_1\ldots i_{m-1}\,k\,i_{m+1}\ldots i_M}).$$

Similarly, a *Kernel Multilinear ICA (K-MICA) Algorithm* results from making the same generalization in the MICA algorithm [Vasilescu & Terzopoulos (2005). Algorithm 2 specifies both K-MPCA and K-MICA. Figure 3(d) unrolls the for-loop in step 3 of Alg. 2.

## 4   INVERSE CAUSAL INFERENCE: MULTILINEAR AND MULTIPLE LINEAR PROJECTIONS

Inverse causal inference estimates the causes of effects, and addresses the why question. Inverse problems often violate one of the conditions of a well-posed problem, and Donald Rubin has referred to the "why" question as "cocktail party chatter" [Gelman & Imbens (2013). For a problem to be well-posed a solution must exit, it must be unique and the solution's behaviour ought to change continuously with the initial conditions [Hadamard (1952). Often, there are multiple combinations of same causal factors that have the same potential outcome. In imaging, these types of outcomes are known as visual illusions.

Therefore, inverse causal inference is the estimation of causes of effects given an estimated forward causal model that is inverted subject to a set of observations that constrain the solution set and render the problem well-posed [Vasilescu et al. (2021)].Similar to reverse causal inference [Gelman & Imbens (2013)], inverse causal inference may be employed as a model checking mechanism and motivation for forward inference question.

---

vectors. Thus, a linear algorithm is easily transformed into a nonlinear algorithm. This trick has been applied to numerous algorithms in machine learning and statistics.

Multilinear projection simultaneously projects one or more unlabeled test images that are not part of the training data set into multiple constituent causal factor spaces associated with data formation, in order to infer the mode labels:

CP or $M$-mode SVD$(\mathcal{T}^{+}{}_{\mathrm{x}} \times_{\mathrm{x}}^{\mathrm{T}} \mathbf{d}_{\mathrm{test}}) \approx \mathbf{r}_1 ... \circ \mathbf{r}_{\mathrm{m}} ... \circ \mathbf{r}_{\mathrm{M}} \circ \mathbf{r}_{\mathrm{E}}.$

Topologically the multilinear projection architecture, Fig. 5, is an inverted M-mode SVD architecture. When the dimensionality of $vec(\mathcal{R})$ is larger than the number of measurements in $\mathbf{d}$, then the system of equations is under determined. There are three possible solutions – dimensionality reduction of the mode matrices, and modeling the mechanism of data formation by multiple linear or tensor models. Instead of performing a multilinear projection, [Vasilescu & Terzopoulos (2002b) perform a set of linear projections.

Figure 5: Neural network architecture of the multilinear projection algorithm [Vasilescu & Terzopoulos (2007) given an estimated interaction causal model, $\mathcal{T}$ ($i.e.$, $\mathbf{T}_{[\mathrm{x}]}$).

## 5 CONCLUSION

This paper introduces deep causal learning architectures that implement tensor factor analysis operations and model the mechanism of data formation. The tensor factor analysis methods, the $M$-mode SVD, the Kernel MPCA/MICA, and the associated causal capsules architectures transform the "cluster" eigenvectors such that the constituent causal representations are invariant of the cluster membership, $i.e.$, invariant of other causal factors of data formation. Causal representation may be computed efficiently by "glomming" together a hierarchy of part-based causal capsules. The hierarchical part-based causal architecture implements the compositional hierarchical tensor factorization, the Incremental M-mode Block SVD. Each part-based capsule analyzes a data column from a hierarchy of data columns [Vasilescu et al. (2021). Inverse causal inference, the estimation of causes of effects, is accomplished with a multilinear projection algorithm. The neural architecture that implements the multilinear projection is an inverted M-mode SVD architecture. Tensor causal factor analysis and their associates neural networks have properties consistent with the capsule theory. Tensor causal factor analysis has been applied on real and synthetic data in many domains, including face recognition where the approach is known as TensorFaces and computer graphics where a set of TensorTextures are synthesized for arbitrary geometries.

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

## A MATHEMATICAL BACKGROUND

Throughout this article, we will denote scalars by lower case italic letters $(a, b, ...)$, vectors by bold lower case letters $(\mathbf{a}, \mathbf{b}, ...)$, matrices by bold uppercase letters $(\mathbf{A}, \mathbf{B}, ...)$, and higher-order tensors by bold uppercase calligraphic letters $(\mathcal{A}, \mathcal{B}, ...)$. Index upper bounds are denoted by italic uppercase letters (*i.e.*, $1 \leq a \leq A$ or $1 \leq i \leq I$). The zero matrix is denoted by $\mathbf{0}$, and the identity matrix is denoted by $\mathbf{I}$. The TensorFaces paper [Vasilescu & Terzopoulos (2002a) is a gentle introduction to tensor factor analysis, [Kolda & Bader (2009) is a great survey of tensor methods and references [Vasilescu (2009); de Lathauwer (1997); Bro (1997) provide an in depth treatment of tensor factor analysis.

### A.1 PCA COMPUTATION WITH LINEAR AUTOENCODER

An autoencoder-decoder that minimizes the reconstruction loss function for a set of observations, $\mathbf{d}_i \in \mathbb{C}^{I_0}$,

$$l \quad = \quad \sum_{i=1}^{I} \|\mathbf{d}_i - \mathbf{B}\mathbf{c}_i\| + \lambda\|\mathbf{B}^{\mathrm{T}}\mathbf{B} - \mathbf{I}\|, \tag{14}$$

and has a linear decoder learns a set of weights, $b_{i_0,r}$, that are identical to the elements of the PCA basis matrix, $\mathbf{B} \in \mathbb{C}^{I_0 \times R}$, when the weights of each neuron are computed sequentially, Fig. 6. An autoencoder is implemented with a cascade of Hebb neurons [Hebb (1949). The contribution of each neuron, $c_1, \ldots, c_r$, is sequentially computed and subtracted from a centered training data set, and the difference is driven through the next Hebb neuron, $c_{r+1}$ [Sejnowski et al. (1989); Sanger (1989); Rumelhart et al. (1986); Ackley et al. (1985); Oja (1982).

The weights of a Hebb neuron, $c_r$, are updated by

$$\Delta\mathbf{b}_r(t+1) = \eta\left(\mathbf{d} - \sum_{i_r=1}^{r}\mathbf{b}_{i_r}(t)c_{i_r}(t)\right)c_r(t) \tag{15}$$

$$= \eta\left(\mathbf{d} - \sum_{i_r=1}^{r}\mathbf{b}_{i_r}(t)\mathbf{b}_{i_r}^{\mathrm{T}}(t)\mathbf{d}\right)\mathbf{d}^{\mathrm{T}}\mathbf{b}_r(t),$$

$$\mathbf{b}_r(t+1) \quad = \frac{(\mathbf{b}_r(t) + \Delta\mathbf{b}_r(t+1))}{\|\mathbf{b}_r(t) + \Delta\mathbf{b}_r(t+1)\|}$$

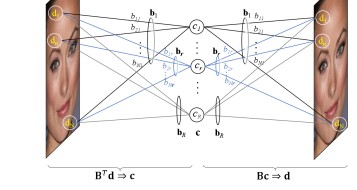

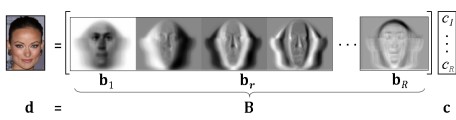

where $\mathbf{d} \in \mathbb{C}^{I_0}$ is a vectorized centered observation with $I_0$ measurements, $\eta$ is the learning rate, $\mathbf{b}_r$ are the autoencoder weights of the $r$ neuron, $c_r$ is the activation, and $t$ is the time iteration. Back-propagation [LeCun et al. (1988; 2012) performs PCA gradient descent [Jolliffe (1986).

Figure 6: Autoencoder-decoder architecture and Principal Component Analysis. (All images have been vectorized, but they are displayed as a grid of numbers. The eigenvector $\mathbf{b}_1$ is the mean and activation $c_1$ is set to 1.)

### A.2 RELEVANT TENSOR ALGEBRA

Briefly, the natural generalization of matrices (i.e., linear operators defined over a vector space), tensors define multilinear operators over a *set* of vector spaces. A *"data tensor"* denotes an $M$-way data array.

**Definition 1 (Tensor)** *Tensors are multilinear mappings over a set of vector spaces, $\mathbb{C}^{I_m}$, $1 \leq m \leq M$, to a range vector space $\mathbb{C}^{I_0}$:*

$$\mathcal{A} : \left\{\mathbb{C}^{I_1} \times \mathbb{C}^{I_2} \times \cdots \times \mathbb{C}^{I_M}\right\} \mapsto \mathbb{C}^{I_0}. \tag{16}$$

*The order of tensor $\mathcal{A} \in \mathbb{C}^{I_0 \times I_1 \times \cdots \times I_M}$ is $M + 1$. An element of $\mathcal{A}$ is denoted as $\mathcal{A}_{i_0 i_1 \ldots i_m \ldots i_M}$ or $a_{i_0 i_1 \ldots i_m \ldots i_M}$, where $1 \leq i_m \leq I_m$.*

The mode-$m$ vectors of an $M$-order tensor $\mathcal{A} \in \mathbb{C}^{I_0 \times I_1 \times \cdots \times I_M}$ are the $I_m$-dimensional vectors obtained from $\mathcal{A}$ by varying index $i_m$ while keeping the other indices fixed. In tensor terminology, column vectors are the mode-0 vectors and row vectors as mode-1 vectors. The mode-$m$ vectors of a tensor are also known as *fibers*. The mode-$m$ vectors are the column vectors of matrix $\mathbf{A}_{[m]}$ that results from *matrixizing* (a.k.a. *flattening*) the tensor $\mathcal{A}$.

**Definition 2 (Mode-$m$ Matrixizing)** *The mode-m matrixizing of tensor $\mathcal{A} \in \mathbb{C}^{I_0 \times I_1 \times \ldots I_M}$ is defined as the matrix $\mathbf{A}_{[m]} \in \mathbb{C}^{I_m \times (I_0 \ldots I_{m-1} I_{m+1} \ldots I_M)}$. As the parenthetical ordering indicates, the*

---

**Algorithm 3** $M$-mode SVD algorithm.

---

**Input** the data tensor $\mathcal{D} \in \mathbb{C}^{I_0 \times \cdots \times I_M}$.
      1. For $m := 0, \ldots, M$,
          Let $\mathbf{U}_m$ be the left orthonormal matrix of $[\mathbf{U_m S_m V_m^T}] := \text{svd}(\mathbf{D}_{[m]})$[a]
      2. Set $\mathcal{Z} := \mathcal{D} \times_0 \mathbf{U_0}^\mathrm{T} \times_1 \mathbf{U_1}^\mathrm{T} \cdots \times_m \mathbf{U}_m^\mathrm{T} \ldots \times_M \mathbf{U}_M^\mathrm{T}$.
**Output** mode matrices $\mathbf{U}_0, \mathbf{U}_1 \ldots, \mathbf{U}_M$, and the core tensor $\mathcal{Z}$.

---

[a] The computation of $\mathbf{U}_m$ in the SVD $\mathbf{D}_{[m]} = \mathbf{U}_m \mathbf{\Sigma} \mathbf{V}_m^\mathrm{T}$ can be performed efficiently, depending on which dimension of $\mathbf{D}_{[m]}$ is smaller, by decomposing either $\mathbf{D}_{[m]} \mathbf{D}_{[m]}^\mathrm{T} = \mathbf{U}_m \mathbf{\Sigma}^2 \mathbf{U}_m^\mathrm{T}$ (note that $\mathbf{V}_m^\mathrm{T} = \mathbf{\Sigma}^+ \mathbf{U}_m^\mathrm{T} \mathbf{D}_{[m]}$) or by decomposing $\mathbf{D}_{[m]}^\mathrm{T} \mathbf{D}_{[m]} = \mathbf{V}_m \mathbf{\Sigma}^2 \mathbf{V}_m^\mathrm{T}$ and then computing $\mathbf{U}_m = \mathbf{D}_{[m]} \mathbf{V}_m \mathbf{\Sigma}^+$.

---

*mode-$m$ column vectors are arranged by sweeping all the other mode indices through their ranges, with smaller mode indexes varying more rapidly than larger ones; thus,*

$$[\mathbf{A}_{[m]}]_{jk} = a_{i_1 \ldots i_m \ldots i_M}, \quad where \tag{17}$$

$$j = i_m \quad and \quad k = 1 + \sum_{\substack{n=0 \\ n \neq m}}^{M} (i_n - 1) \prod_{\substack{l=0 \\ l \neq m}}^{n-1} I_l.$$

A generalization of the product of two matrices is the product of a tensor and a matrix [de Lathauwer (1997); Carroll et al. (1980).

**Definition 3 (Mode-$m$ Product, $\times_\mathbf{m}$)** *The mode-$m$ product of a tensor $\mathcal{A} \in \mathbb{C}^{I_1 \times I_2 \times \cdots \times I_m \times \cdots \times I_M}$ and a matrix $\mathbf{B} \in \mathbb{C}^{J_m \times I_m}$, denoted by $\mathcal{A} \times_m \mathbf{B}$, is a tensor of dimensionality $\mathbb{C}^{I_1 \times \cdots \times I_{m-1} \times J_m \times I_{m+1} \times \cdots \times I_M}$ whose entries are computed by*

$$[\mathcal{A} \times_m \mathbf{B}]_{i_1 \ldots i_{m-1} j_m i_{m+1} \ldots i_M} = \sum_{i_m} a_{i_1 \ldots i_{m-1} i_m i_{m+1} \ldots i_M} b_{j_m i_m},$$

$$\mathcal{C} = \mathcal{A} \times_m \mathbf{B}. \quad \underset{\text{tensorize}}{\overset{\text{matrixize}}{\longleftrightarrow}} \quad \mathbf{C}_{[m]} = \mathbf{B} \mathbf{A}_{[m]}.$$

The $M$-mode SVD, Alg. 1 [Vasilescu & Terzopoulos (2002a) is a "generalization" of the conventional matrix (i.e., 2-mode) SVD which may be written in tensor notation as

$$\mathbf{D} = \mathbf{U}_0 \mathbf{S} \mathbf{U}_1^\mathrm{T} \quad \Leftrightarrow \quad \mathbf{D} = \mathbf{S} \times_0 \mathbf{U}_0 \times_1 \mathbf{U}_1$$

The $M$-mode SVD orthogonalizes the $M$ spaces and decomposes a tensor as the *mode-$m$ product*, denoted $\times_m$, of $M$-orthonormal mode matrices, and a core tensor $\mathcal{Z}$

$$\mathcal{D} = \mathcal{Z} \times_0 \mathbf{U}_0 \cdots \times_m \mathbf{U}_m \cdots \times_M \mathbf{U}_M. \tag{18}$$

$$\mathbf{D}_{[m]} = \mathbf{U_m} \mathbf{Z}_{[m]} (\mathbf{U}_M \cdots \otimes \mathbf{U_{m+1}} \otimes_{\text{m-1}} \mathbf{U} \cdots \otimes \mathbf{U}_0)^\mathrm{T}, \tag{19}$$

$$vec(\mathcal{D}) = (\mathbf{U}_M \cdots \otimes \mathbf{U_{m+1}} \otimes \mathbf{U_{m-1}} \cdots \otimes \mathbf{U}_0) \, vec(\mathcal{Z}). \tag{20}$$

The latter two equations express the decomposition in matrix form and in terms of $vec$ operators.

### A.3 COMPOSITIONAL HIERARCHICAL BLOCK TENSORFACES

Training Data: In our experiments, we employed gray-level facial training images rendered from 3D scans of 100 subjects. The scans were recorded using a CyberwareTM 3030PS laser scanner and are part of the 3D morphable faces database created at the University of Freiburg [Blanz & Vetter (1999). Each subject was combinatoriall y imaged in Maya from 15 different viewpoints ($\theta = -60°$ to $+60°$ in $10°$ steps on the horizontal plane, $\phi = 0°$) with 15 different illuminations ($\theta = -35°$ to $+35°$ in $5°$ increments on a plane inclined at $\phi = 45°$).

Data Preprocessing: Facial images were warped to an average face template by a piecewise affine transformation given a set of facial landmarks obtained by employing Dlib software [King (2009); Kazemi & Sullivan (2014); Si et al. (2013); Macedo et al. (2006); Hatamizadeh et al. (2019). Illumination was normalized with an adaptive contrast histogram equalization algorithm, but rather than performing contrast correction on the entire image, subtiles of the image were contrast normalized, and tiling artifacts were eliminated through interpolation. Histogram clipping was employed to avoid over-saturated regions.

Experiments: Each image, $\mathbf{d} \in \mathbb{R}^{I_0 \times 1}$, was convolved with five filters banks $\{\mathbf{H}_s \| s = 1...S\}$. The filtered images, $\mathbf{d} \times_0 \mathbf{H}_s$, resulted in five facial part hierarchies composed of (i) independent pixel parts (ii) parts segmented from different layers of a Gaussian pyramid that were equally or (iii) unequally weighed, (iv) parts were segmented from a Laplacian pyramid that were equally or (v) unequally weighed. We ran five experiments with five facial part hierarchies from which a person representation was computed, Fig. 8. The composite person signature was computed for every test image by employing the multilinear projection algorithm [Vasilescu (2011); Vasilescu & Terzopoulos (2007), and signatures were compared with a nearest neighbor classifier.

To validate the effectiveness of our system on real-world images, we report results on "LFW" dataset (*LFW*) [Huang et al. (2007). This dataset contains 13,233 facial images of 5,749 people. The photos are unconstrained (*i.e.*, "in the wild"), and include variation due to pose, illumination, expression, and occlusion. The dataset consists of 10 train/test splits of the data. We report the mean accuracy and standard deviation across all splits in Table 2. Fig. 8(b-c) depicts the experimental ROC curves. We follow the supervised *"Unrestricted, labeled outside data"* paradigm.

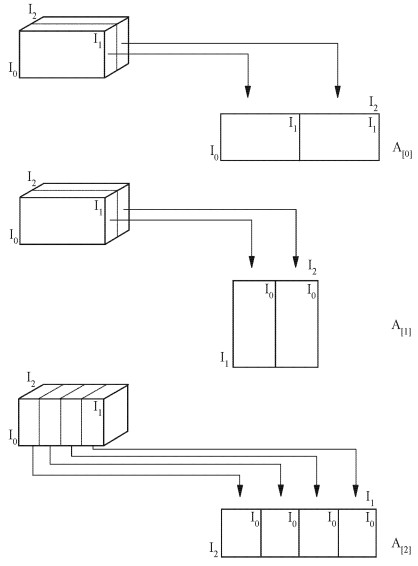

Figure 7: Matrixizing a $3^{\text{rd}}$ order tensor, $\mathcal{A}$. The tensor can be matrixized in 3 ways.

Results: While we cannot celebrate closing the gap on human performance, our results are promising. DeepFace, a CNN model, improved the prior art verification rates on LFW from $70\%$ to $97.35\%$, by training on $4.4M$ images of $200 \times 200$ pixels from $4,030$ people, the same order of magnitude as the number of people in the LFW database.

We trained on less than one percent $(1\%)$ of the 4.4M total images used to train DeepFace. Images were rendered from 3D scans of 100 subjects with an the intraocular distance of approximately 20 pixels and with a facial region captured by $10,414$ pixels (image size $\approx 100 \times 100$ pixels). We have currently achieved verification rates just shy of $80\%$ on LFW. When data is limited, CNN models do not convergence or generalize.

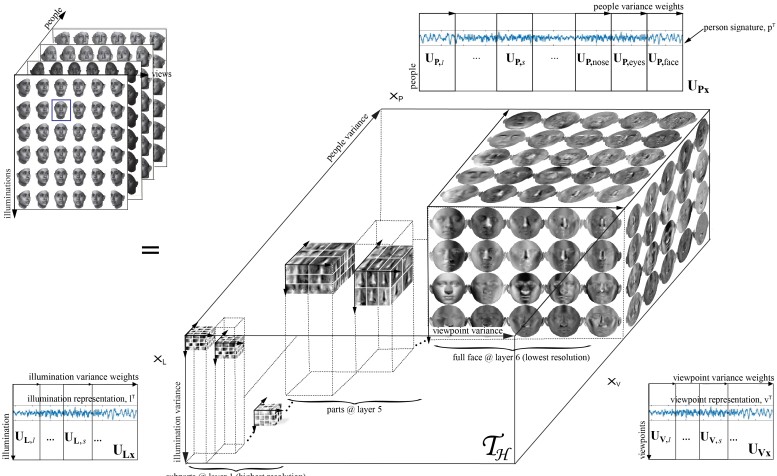

Figure 8: Compositional hierarchical Block TensorFaces learns a hierarchy of features, and reesents each person as a part-based compositional representation. Figure depicts the training data factorization, $\mathcal{D} = \mathcal{T}_{\mathcal{H}} \times_{\text{L}} \mathbf{U}_{\text{L}} \times_{\text{V}} \mathbf{U}_{\text{V}} \times_{\text{P}} \mathbf{U}_{\text{P}}$, where an observation is represented as $\mathbf{d}(\mathbf{p}, \mathbf{v}, \mathbf{l}) = \mathcal{T}_{\mathcal{H}} \times_{\text{L}} \mathbf{l} \times_{\text{V}} \mathbf{v} \times_{\text{P}} \mathbf{p}$ and $\mathcal{T}_{\mathcal{H}}$ spans the *hi*erarchical causal factor variance.

| Test Dataset | PCA | TensorFaces | compositional hierarchical Block TensorFaces | | | | |
|---|---|---|---|---|---|---|---|
| | | | Pixels | Gaussian Pyramid | Weighted Gaussian Pyramid | Laplacian Pyramid | Weighted Laplacian Pyramid |
| Freiburg | 65.23% | 71.64% | 90.50% | 88.17% | 94.17% | 90.96% | 93.98% |
| LFW | 69.23% ±1.51 | 66.25% ±1.60 | 72.72% ±2.14 | 76.72% ±1.65 | 77.85% ±1.83 | 77.58% ±1.45 | 78.93% ±1.77 |

Table 2: Empirical results reported for LFW : PCA, TensorFaces and compositional hierarchical Block TensorFaces. *Pixels* denotes independent facial part analysis *Gaussian/Laplacian* use a multi resolution pyramid to analyze facial features at different scales. *Weighted* denotes a weighted composite signature.

Freiburg Experiment:

Train on Freiburg: 6 views ($\pm60°,\pm30°,\pm5°$); 6 illuminations ($\pm60°,\pm30°,\pm5°$), 45 people
Test on Freiburg:   9 views ($\pm50°, \pm40°, \pm20°, \pm10°, 0°$), 9 illums ($\pm50°, \pm40°, \pm20°, \pm10°, 0°$), 45 different people
LFW Experiment: Models were trained on approximately half of one percent ($0.5\% < 1\%$) of the 4.4M images used to train DeepFace.
Train on Freiburg:
15 views ($\pm60°,\pm50°, \pm40°,\pm30°, \pm20°, \pm10°,\pm5°, 0°$), 15 illuminations ($\pm60°,\pm50°, \pm40°,\pm30°, \pm20°, \pm10°,\pm5°, 0°$), 100 people
Test on LFW: We report the mean accuracy and standard deviation across standard literature partitions [Huang et al. (2007)], following the
*Unrestricted, labeled outside data* supervised protocol.

**Summary:** This paper contributes to the tensor algebraic paradigm and models cause-and-effect as a hierarchical block tensor interaction between intrinsic and extrinsic hierarchical causal factors of data formation.

A data tensor expressed as a function of a hierarchical data tensor is a unified tensor model of wholes and parts from which a new compositional hierarchical block tensor factorization was derived. The resulting causal factor representations are interpretable, hierarchical, and statistically invariant to all other causal factors. Our approach was demonstrated in the context of facial images by training on a very small set of synthetic images. While we have not closed the gap on human performance, we report encouraging face verification results on two test data sets–the Freiburg, and the Labeled Faces in the Wild datasets. CNN verification rates improved the $70\%$ prior art to $97.35\%$ when they employed 4.4M images from $4,030$ people, the same order of magnitude as the number of people in the LFW database. We have currently achieved verification rates just shy of $80\%$ on LFW by employing synthetic images from 100 people for a total of less than one percent ($1\%$) of the total images employed by DeepFace. By comparison, when data is limited, CNN models do not convergence, or generalize.

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
