# OpenReview forum: "Neural network architectures for disentangling the multimodal structure of data ensembles"
_ICLR.cc/2022/Conference — ICLR 2022 Submitted_

### Official Review · Reviewer_oC53 · 2021-10-29

**Correctness:** 3
**Technical Novelty And Significance:** 2
**Empirical Novelty And Significance:** 2
**Recommendation:** 3
**Confidence:** 3

**Main Review:**

Strengths:
* the paper adresses an important and timely problem: that of causal discovery
* the paper proposes an innovative approach to combine Tensor Factor Analysis with capsule network architecture that uses auto-encoding architectures

Weaknesses:
* This paper suffers from a major lack of clarity in writing. Here are some important points
- The introduction essentially lists a number of papers that use Tensor-Decomposition for various applications but does not clearly identify a gap in knowledge or a problem that will be addressed by the contribution. It states that the paper seeks to "connect tensor causal factor analysis with deep learning" but does not adequately articulate why this is needed or a good idea.
- There is a lack of clarity in definitions: (i) several acronyms are not defineds (e.g. MPCA/MICA). (ii) in section 2, equation (1), there is no desciption of the variables (what is d, B, c ?) (iii) Throughout all figures appear to show image recognition/classification tasks but nowhere is the task or computation described. (iv) formatting is not at the level of publications (Figures are not referred in the text in the order they appear, a number of citations are compiled with an error showing "?", punctuation and sentence construction is as mishandled in a number of areas).

*Most crucial is the lack of experimental evidence. The figures appear to show some image recognition task that is never described and there is no comparison with other methods, nor any description of the advantages of the proposed method.

*Conclusion is quasi non-existant and there is no discussion about future directions for this work and how it may be improved in the future.


**Summary Of The Paper:**

The authors present a framework in which Tensor Factor Analysis is incorporated in a causal graph structure implemented by a series of "capsule" auto-encoders.  The rationale is that these capsules can be combined in some way to create meaningful hierarchical architecture useful for causal inference.

**Summary Of The Review:**

In my opinion, this paper is not clearly written, and does not contain compelling and convincing evidence that the proposed method adresses a gap in knowledge or improves on current method. It suffers from major structural problems as outlined in detail above. It is quite possible that the contribution is, in fact, significant but I could not evaluate it because of these issues.

---

### Official Review · Reviewer_Bzxf · 2021-10-31

**Correctness:** 3
**Technical Novelty And Significance:** 3
**Empirical Novelty And Significance:** Not applicable
**Recommendation:** 5
**Confidence:** 3

**Main Review:**

Pros:
1. As far as I understand, the authors provide a causal invariant representation of the data in a subspace. The representation can be used for streaming tasks.
2. The cascade idea depicted in Fig. 5 is neat!
3. The extension to non-linear Causal capsules using kernels is intuitive and nicely introduced.

Concerns & questions:
1. The paper is confusing to read. The motivation and the flow of the paper is tedious to follow. This might be due to the lack of knowledge on my end. Although, I am fairly familiar with causality theory (by Pearl etc.), capsule networks and tensor decompositions. The authors should provide more background in their paper.
2. It will be great if the authors can include some set of experiments to demonstrate the benefits of their approach in practice.
3. I can foresee some issues with the scalability of this work. I request the authors to highlight them in the paper. This will be a good indicator of how useful this technique can be for real world applications.
4. Missing citations on Pg. 2 and some minor typos.


**Summary Of The Paper:**

This paper is an attempt to connect tensor factor analysis with DNN learning. The paper explores the forward causal inference and inverse causal inference problems. The forward causal inference is performed using causal capsules architecture that uses M-mode SVD for subspace learning (autoencoder). The authors show that the M-mode SVD rotates the basis vectors of various clusters such that the causal representation is the same regardless of the data cluster (Fig.1). They introduce causal capsule hierarchy architecture that uses tensor decompositions for forward and inverse causal inference.

**Summary Of The Review:**

I would be more confident about my review of the work, if the authors can include some set of experiments in the paper. It will be very helpful if the authors can point out the benefits over the existing SOTA techniques on why to use their proposed causal representation. Some comments on the scalability of their approach will also be appreciated.

I am willing to reconsider my ratings if the authors address some of my concerns.

---

### Official Review · Reviewer_4HKq · 2021-11-01

**Correctness:** 1
**Technical Novelty And Significance:** 1
**Empirical Novelty And Significance:** 1
**Recommendation:** 3
**Confidence:** 3

**Main Review:**

My main concern with this paper is the effectiveness of the proposed method. Since there is no experimental evaluation, I don’t know how well the proposed method will perform in real problems. Also, if this paper claims a theoretical contribution, there should be strong theoretical justifications about the advances of the proposed method. However, I only see the method descriptions instead of in-depth theoretical analyzes. Besides, this paper is very hard to follow. It is not well structured and has many typos.


**Summary Of The Paper:**

This paper tried to build the connection between tensor factorization and capsule network, and a forward causal inference with causal capsule method was proposed. However, since there is no theoretical justification or experimental evaluation, I cannot judge the effectiveness of the proposed method.

**Summary Of The Review:**

I think this paper's contributions are not well justified. Given its current status, I cannot vote for its acceptance.

---

### Official Review · Reviewer_GQkp · 2021-11-03

**Correctness:** 2
**Technical Novelty And Significance:** 2
**Empirical Novelty And Significance:** Not applicable
**Recommendation:** 3
**Confidence:** 2

**Main Review:**

This paper is hard to read. Even though I read multiple times, I'm not sure if I fully understand this paper. The idea to connect tensor analysis with casual inferences is good but this paper doesn't have enough contexts.

Some issues:

- There are a few typos and incorrect latex references that impacts the reading experiences. For example:
  - In the second paragraph in page 2, "by proposing a causal capsule hierarchy of parts and wholes that is consistent the capsule theory" should be "consistent with the capsule thoery".
  - In the second last paragraph in page 2, there are two "?" in the citations after "which may result in t
approximate subspace".
  - Right above section 3.2 in page 8, there is a "Fig ??" after "not just a complete observation,".
- It would be better to provide some empricial results to show the effectness of the proposed models. The face recognitions task is used as an example in many places in the paper, but there are no experiments to evaluate how the proposed model performs.
- It's worth to discuss what kind of tensors are suitable for the proposed model. It's also worth to show the proposed model can run in a reasonable speed for modern datasets.

**Summary Of The Paper:**

The authors present some analysis and describes a new methodology on how to do forward/inverse causal inferences with tensor analysis. There are no empricial results nor theoratic analysis to back the authors' claims.

**Summary Of The Review:**

 I don't think this paper is ready to be read. I think there are many places that can be improved.

---

### Decision · Program_Chairs · 2022-01-20

**Decision:**

Reject

**Comment:**

Four experts reviewed this paper and rated the paper below the acceptance threshold. The reviewers raised many concerns regarding the paper, mainly the lack of empirical studies and clarity. Some reviewers also suggested the authors better positioning the paper in the literature by discussing more related works. The rebuttal did not address all concerns. Considering the reviewers' concerns, we regret that the paper cannot be recommended for acceptance at this time. The authors are encouraged to consider the reviewers' comments when revising the paper for submission elsewhere.

---

> ### Public Comment · ~M._Alex_O._Vasilescu1 · 2022-02-12
> **Assessment of the paper decision**
>
> > Four experts reviewed this paper
>
> To recap, this paper was reviewed by four people -- one of which had difficulty with reading comprehension, but did not specify what specifically was the issue.  The rest of the reviewer requests were outside the scope of this paper.
>
> This paper provides a closed-form derivation that translated tensor factor analysis into a causal deep neural network architecture.  The supplemental contains empirical results despite being outside of the scope of this paper.
>
> One reviewer requested that I add a set of references to literature that is not directly related to this body of work. I would have complied but the reviewer modified one of their old comments with a set of references after the deadline to upload a new version of the paper.  This gives the incorrect appearance that I have not responded to their concern.
>
> Reviewers should not be allowed to substantively change old comments with new content after a new paper is uploaded.
>
> > The authors are encouraged to consider the reviewers' comments when revising the paper for submission elsewhere.
>
> I will take your suggestion to submit my paper "elsewhere" under advisement.